# Human–Computer Interaction in Healthcare: A Bibliometric Analysis with CiteSpace

**DOI:** 10.3390/healthcare12232467

**Published:** 2024-12-06

**Authors:** Xiangying Zhao, Shunan Zhang, Dongyan Nan, Jiali Han, Jang Hyun Kim

**Affiliations:** 1Department of Interaction Science, Sungkyunkwan University, Seoul 03063, Republic of Korea; zxy94@g.skku.edu (X.Z.); 970205@g.skku.edu (S.Z.); 2Department of Human-Artificial Intelligence Interaction, Sungkyunkwan University, Seoul 03063, Republic of Korea; 3School of Business, Macau University of Science and Technology, Macau 999078, China; dynan@must.edu.mo; 4Department of Epidemiology, Richard M. Fairbanks School of Public Health, Indiana University Indianapolis, Indianapolis, IN 46202, USA; jialhan@iu.edu

**Keywords:** healthcare, HCI, bibliometric analysis, CiteSpace

## Abstract

Background/Objectives: Studies on the application and exploration of human–computer interaction (HCI) technologies within the healthcare sector have rapidly expanded, showcasing the immense potential of HCI to enhance medical services, elevate patient experiences, and advance health management. Despite this proliferating interest, there is a notable shortage of comprehensive bibliometric analyses dedicated to the application of HCI in healthcare, which limits a thorough comprehension of the growth trends and future trajectories in this area. Methods: To bridge this gap, we employed bibliometric methods using the CiteSpace tool to systematically review and analyze the current state and trends of HCI research in healthcare. A meticulous topic search of Web of Science yielded 3598 papers published between 2004 and 2023. Results: Through literature analysis, the most productive researchers, institutes, and countries/territories and the collaboration networks among authors and countries within the field were analyzed. Additionally, by conducting a co-citation analysis, journals and literature with high citation rates and influence within the academic community in this field were revealed. Through a cluster analysis based on literature co-citations and keyword burst analyses, we further explored the main research themes and hot topics within the fields of healthcare and HCI. Conclusions: In summary, through a comprehensive and systematic bibliometric analysis, this study provides a solid knowledge foundation for HCI in the healthcare research community, thereby fostering the development of innovative research and the optimization of practical applications in the field.

## 1. Introduction

Human–computer interaction (HCI) comes from the commitment to design computer systems that enhance human activity and ensure efficient and secure operations [1]. HCI is fundamental to the usability and effectiveness of technological innovations, including robotic systems and software tools that support healthcare professionals and patients [2,3,4,5]. Recent developments have underlined the substantial impact of HCI on technology adoption in healthcare, shaping user experiences for nurses and specialized groups [2]. The advent of technology in the medical context—from robotic surgical systems to electronic health records—highlights the growing interdependence between healthcare delivery and technological advancements [6].

Amid this evolving landscape, HCI in healthcare has emerged as an increasingly popular research focus. However, rare studies are conducting bibliometric analysis in the HCI healthcare area. Bibliometric analysis can objectively evaluate performance and map science across areas [7], which helps researchers and stakeholders easily understand a particular academic area [8]. For instance, a bibliometric analysis tailored to HCI applications in healthcare is expected to provide some insights into the development patterns in the area, aiding practitioners and policymakers in harnessing HCI interventions to enhance patient and caregiver well-being. Consequently, our research performed a bibliometric analysis of HCI healthcare to offer a comprehensive understanding of it. In accordance with the recommendations of previous studies [9,10,11], we conducted the analyses. Our multifaceted approach encompassed the following:Identification of prolific researchers, institutions, and nations/regions;Analysis of academic collaboration networks between researchers or countries/regions;Co-citation analysis at journal and reference levels;Cluster analysis of reference co-citations to discern primary research themes;Citation burst analysis to track rising study subjects.

Consequently, we proposed the following research questions (RQs):RQ1:Which researchers, institutions, and countries/regions are the leading contributors in the field of HCI in healthcare?RQ2:What is the status of academic collaboration among researchers or countries/regions in the field of HCI in healthcare?RQ3:What are the most influential journals and studies in the field of HCI in healthcare?RQ4:What are the main themes in the field of HCI in healthcare?RQ5:What are the emerging trends in the field of HCI in healthcare?

These research questions helped identify shared cognitive orientations among scholars. Furthermore, the social connections revealed by the networks explored in this study enhanced our understanding of scholarly works, scholars, their affiliations, and countries/regions. This type of multilevel analysis is a systematic academic research method.

## 2. Method

### 2.1. Data Collection

Web of Science (WoS) is regarded as a key resource for bibliometric analyses in scientific research [12,13,14,15,16]. As suggested by Damar et al. [17] and Abbas et al. [9], in this study, WoS was recognized as the origin of HCI healthcare-associated literature data. 

The primary data sources consulted for this research were the Science Citation Index Expanded (SCI-EXPANDED), 1990–present; Social Sciences Citation Index (SSCI), 1983–present; Arts & Humanities Citation Index (AHCI), 1983–present; Conference Proceedings Citation Index—Science (CPCI-S), 1990–present; Conference Proceedings Citation Index—Social Science & Humanities (CPCI-SSH), 1990–present; and Emerging Sources Citation Index (ESCI), 2019–present; provided by the WoS databases. The data were collected on 3 March 2024. 

This study selected keywords to extract bibliometric data, considering Guo et al. [18] and the guidance of professors in the field of HCI and healthcare (see Table 1). These terms were found in the titles, abstracts, or keywords of the following English-language document types: “article”, “proceeding paper”, and “review article”. Kumar et al. [19] reported that data spanning a full year should be assessed to provide accurate bibliometric analysis outcomes. Thus, only articles published from 2004 to 2023 (20 years) were included, while those published in 2024 were excluded. Table 1 summarizes the data extraction process used in this study. 

### 2.2. Data Analysis

There are several instruments such as CiteSpace, VOSviewer, BibExcel, HistCite, Pajek, Netdraw, and Publish or Perish that have been widely employed for bibliometric studies [20,21,22,23,24,25]. In our research, CiteSpace [20], developed by Professor Chaomei Chen, was selected. Comparing tools such as VOSviewer, BibExcel, and HistCite, Citespace has more functions required by scientometric studies, such as dual-map overlay, automatic cluster naming, and citation burst analysis [8,22,26].

Specifically, the CiteSpace software package was used to obtain collaboration networks (e.g., author and country/region-based co-authorship networks) and influence networks (e.g., co-cited reference and journal-based networks) to conduct cluster and keyword burst analyses. CiteSpace is a prominent tool in visual analytics [9], whereby a field of study can be explored by identifying hotspots and academic trends using a combination of systematic mapping and bibliometric analyses, scientometrics, and visual–analytic methodologies [27]. The primary aim was to accentuate the structure and evolution of a particular domain [20]. Therefore, this study employed the recent version of CiteSpace, CiteSpace 6.3. R1 (64-bit) Advanced.

## 3. Outcomes and Discussion

### 3.1. Annual Publication Volume

Figure 1 reveals the annual number of publications in the HCI healthcare area. The growth of publications was relatively slow in the early years but began to accelerate around 2015, with a notable enhancement starting in 2019. The reasons for this growth trend are as follows: First, the expansion of the WoS database and the inclusion of ESCI, which led to the addition of more journals [15]. Second, the outbreak of the COVID-19 pandemic in 2019. Third, the application of HCI technology across various fields such as healthcare, which has become increasingly common. 

### 3.2. Prolific Researchers, Institutions, and Countries/Territories

The primary driving forces in a particular field are determined by identifying prolific authors/researchers, institutions, and countries/territories [8]. Therefore, our research also explored the main research forces regarding HCI in the healthcare domain at the abovementioned levels. 

#### 3.2.1. Prolific Authors and Their Co-Authorship Networks

Table 2 lists the 10 most prolific authors, ranked by their number of published papers. The top three authors were Elizabeth Broadbent, with 20 published papers; Madeline Balaam, with 10; and Ho Seok Ahn, with 9. These findings suggest that these three authors are the most prolific researchers in the discipline of HCI in healthcare. These were followed by Goldie Nejat with eight articles; Bruce MacDonald, Gavin Doherty, and Eduard Fosch-Villaronga with seven articles each; and Norina Gasteiger, Suzanne Bakken, and Dena Al-Thani with six articles each.

Furthermore, in our author co-authorship analysis, which culminated in the visual representations shown in Figure 2, we mapped the collaborative landscape of the HCI–healthcare research community. The nodes within this network signify authors, with the size of each node correlating to the author’s publication output; larger nodes represent a larger number of published works. The ties between these nodes depict co-authorship links; the thickness of the lines illustrates the frequency of collaboration. Figure 1 shows that the author-based collaboration network in the HCI healthcare domain consists of several disjointed subnetworks.

Additionally, we discovered a large subnetwork consisting of Elizabeth Broadbent, Ho Seok Ahn, and Bruce MacDonald, among others (Figure 3). At the heart of the network is Elizabeth Broadbent, a major node with multiple connections and the most published author in her/his field. Her/his collaborations with various researchers in the field also reflect her/his key role as an important contributor. Networks reveal not only individual productivity but also the power of collaborative efforts. Also, Elizabeth Broadbent’s research group has focused on interactions between humans and healthcare robots [28,29], a subject with substantial potential to shape future research directions in the HCI–healthcare academic area.

#### 3.2.2. Prolific Institutions

An institution analysis revealed that the University of Washington stands at the forefront, with an impressive number of 71 published articles, leading the field of healthcare-related HCI. Closely following are the Chinese Academy of Sciences with 69 publications and the Georgia Institute of Technology with 41 publications, both of which demonstrate substantial contributions to the field. Other notable institutions include the University College London and Tsinghua University, which have significantly influenced the domain with 39 and 33 articles, respectively. 

Table 3 provides a detailed ranking of the top ten institutions, based on their publication output in healthcare-related HCI, highlighting their pivotal roles in advancing research within this crucial area of study. This ranking not only reflects the academic output of these institutions but also underscores their leading positions in facilitating cutting-edge research and development in the HCI healthcare sector.

#### 3.2.3. Prolific Countries/Regions and Their Collaboration Networks 

The countries/regions ranked in the top ten, based on the number of published papers, are presented in Table 4: USA (945 papers), People’s Republic of China (721 papers), England (322 papers), Germany (232), Australia (177), India (176), Canada (175), Italy (146), Spain (116), and the Netherlands (114). These data not only indicate the sheer volume of research output but also offer insights into countries’ varying levels of investment and activity in the burgeoning field of HCI in healthcare.

The dominance of the United States in HCI–healthcare research is clear, with a substantial lead in the number of published papers. Following closely behind, the People’s Republic of China emerged as a formidable contender in the HCI–healthcare field. With a significant number of published papers, China’s rapid increase in publications underscores its increasing investment in research and development within the healthcare sector. England and Germany are also prominent contributors to healthcare research on HCI, with varying degrees of output. Furthermore, the inclusion of Australia, India, Canada, Italy, Spain, and the Netherlands among the top-ranking countries highlights widespread interest and engagement in healthcare research on HCI across diverse geographical regions. Each of these countries has unique perspectives, expertise, and research priorities, enriching the global discourse on HCI applications in healthcare.

We also conducted a country/region-based coauthorship network analysis and generated a collaboration network, as shown in Figure 4. Nodes represent countries/territories; the link between nodes indicates cooperation between countries/territories. The node size indicates the number of publications in a corresponding country/region. The pink node edges indicate that the node has a strong betweenness centrality. Betweenness centrality is “the number of times a node lies on the shortest path between other nodes” [30]. 

The main findings are as follows. First, the USA has the highest degree of betweenness centrality, which means that it plays the most important role in the entire collaboration network; this is consistent with them being the top three countries/territories in the world with the most publications in this field In addition, the publication volume of France, Saudi Arabia, Pakistan, Spain, and the United Arab Emirates is relatively small, but their betweenness centrality is relatively high (see Table 5).

### 3.3. Co-Citation Analysis

Chang et al. [31] reported that when two or more papers or journals are simultaneously cited by a third paper, a co-citation relationship is established. Several scholars [27,32] have implied that co-citation analysis is one of the most widely employed methods for identifying impactable journals or papers/references in a particular academic area. In general, the most co-cited references/papers become hotspots, reflecting the knowledge base and academic trends [27]. In this study, a co-citation analysis was performed at the journal and literature levels within the field of HCI in healthcare.

#### 3.3.1. Journal Co-Citation Analysis

We employed CiteSpace to perform a journal co-citation analysis. The resulting table (Table 6) displays the top 10 journals ranked by citation count, reflecting their influence and interconnectedness within the field. *Lecture Notes in Computer Science* secured the top position with 758 citations, establishing itself as a pivotal source in the field. *Sensors* was cited 549 times, holding a significant position in scholarly discourse in this field. *PLoS ONE* ranked third with 488 citations, whereas the *Journal of Medical Internet Research* was cited 452 times, securing the fourth position. These are noteworthy sources of information that are important in the research domain of HCI in healthcare.

#### 3.3.2. Reference Co-Citation Analysis

Figure 5 shows eight papers that received at least 21 citations in the HCI-in-healthcare co-citation network. In this case, a node represents a literature article and the line connecting two nodes represents the co-citation relationship between the two articles. Notably, an article’s citation count increases with the node size. A short distance between two nodes suggests that the articles are frequently co-cited.

Table 7 lists the top eight highly cited references with a strong impact on the field of HCI in healthcare. Specifically, the paper by Yang et al. [33] was cited 31 times, followed by papers written by Hua et al. [34], Trung and Lee [35], Cheng et al. [36], and Abdi et al. [37] with 26, 23, 23, and 23 citations, respectively.

This literature was selected as the top eight articles on HCI in the healthcare field, reflecting the importance and influence of electronic skin technology in HCI. First, these articles focused on the advances in electronic skin technology, including key technologies in specific application areas, such as skin-attached devices, robotics, and prosthetics. This demonstrates the diversity and potential of electronic skin technologies in healthcare. 

Second, these studies covered different types of electronic skin sensor designs and preparation methods, thereby providing diversity and breadth for future research and applications. Therefore, the importance of these articles lies not only in their innovation and contribution but also in charting the development direction of electronic skin technology in the field of HCI in healthcare. They provide a valuable reference and guidance for future research and applications. 

#### 3.3.3. Cluster Analysis Based on Literature Co-Citations

We performed a cluster analysis on a reference co-citation network to investigate the main themes of HCI in the healthcare field and found nine main clusters (shown in Figure 6). To achieve a more objective and detailed insight into the key themes, we employed a cluster analysis with the log-likelihood ratio (LLR) method to generate labels for these clusters [8,41,42,43]. The LLR clustering method can produce high-quality clusters characterized by significant intraclass variability, minimal interclass similarity, and comparability between classes [41].

The top five clusters are listed in Table 8. Cluster #0 (interlocked carbon nanotube array) was considered the most important cluster in the field of healthcare HCI, followed by Cluster #1 (triboelectric nanogenerator), Cluster #2 (patient-generated data), Cluster #3 (wearable strain sensor), and Cluster #5 (women’s health). The reasons for these findings are twofold. First, all these clusters exhibited silhouette values exceeding 0.8, indicating a strong fit [8,43,44]. Second, these clusters were among the five largest clusters in the analysis.

Cluster #0, labeled “interlocked carbon nanotube array”, was the largest cluster within this study, with 117 members and a high silhouette value of 0.943, indicating tight clustering. This cluster primarily explores nanotechnology, sensing technologies, and electrochemical analysis. In the main cited article within this cluster, Shi et al. [46] extensively reviewed various morphological structures in pressure sensors and their advanced sensing capabilities. They also discussed intelligent pressure sensor applications and related manufacturing technologies. 

Cluster #1, labeled “triboelectric nanogenerator”, was the second largest cluster, with 68 members and a silhouette value of 0.922, reflecting strong cohesion. This cluster explores triboelectric nanogenerators, human–machine interactions, and their versatile applications. The key article by He et al. [47] reviewed advances in flexible microstructural pressure sensors for healthcare and human–machine applications, emphasizing the design and materials that enhance sensor performance and outlining future challenges and prospects.

Cluster #2, labeled “patient-generated data”, was the third largest group, with 55 members and a high silhouette value of 0.99, indicating very cohesive research topics. This cluster primarily focuses on patient-generated data and the perspectives of smokers. The central article by Mishra et al. [48] examined how hospitalized patients track their health data and interact with their care teams. This study used a stage-based personal informatics model to explore patient needs for collaborative health tracking in hospital settings and suggested modifications to support this process.

Cluster #3, labeled “wearable strain sensor”, was the fourth largest group, with 40 members and a silhouette value of 0.968, indicating high coherence in research topics. This cluster focused on wearable and flexible strain sensors, particularly those utilizing nanofiber yarn. The key article by Gao et al. [49] reviewed advancements in wearable strain sensors made from electrospun fibers, emphasizing their role in personalized healthcare. They discussed the fabrication, properties, and performance optimization of these sensors, as well as their applications in biomonitoring, motion detection, and human–machine interaction.

Cluster #5, labeled “women’s health”, comprised 35 members with a silhouette value of 0.917, indicating strong thematic coherence. It centers on women’s health and HCI gender issues. The key article by Tuli et al. [50] investigated menstrual health education in India, revealing gaps between educational practices and societal expectations, and discussed the sociotechnical ramifications for the design of menstrual health education programs using a feminist HCI lens.

#### 3.3.4. Keyword Analysis with Citation Bursts

The keywords were the core word extractions offered by scholars [18]. Figure 7 shows the 20 keywords with the strongest citation bursts. The main findings of the keyword burst analysis are as follows: First, topics such as patient safety, electronic health records, and big data can be considered hot topics in the field of HCI in healthcare from 2004 to 2023. Second, artificial intelligence, deep learning, and machine learning have been the hottest topics in the past three years, which means that HCI–healthcare research tends to focus on AI and its applications.

## 4. Conclusions

In this study, we meticulously charted the landscape of HCI within the healthcare sector through an extensive bibliometric analysis. Our findings highlight the prolific contributions of scholars such as Elizabeth Broadbent, Madeline Balaam, and Ho Seok Ahn, as well as the network of key authors associated with Elizabeth Broadbent. Institutions such as the University of Washington, the Chinese Academy of Sciences, and the Georgia Institute of Technology are major contributors, reflecting the large amount of research activities carried out by these academic institutions.

The United States, the People’s Republic of China, England, Germany, and Australia were identified as the most productive countries/territories in terms of publication volume. This suggests that these countries/territories also have more funding for investments in HCI in healthcare. In particular, the United States and Chinese governments have invested heavily in healthcare and computing technology, making significant contributions to HCI–healthcare research [51,52,53], which means that external factors (e.g., government support) can also influence the research landscape, not just individual author performance [54]. However, the centrality within the co-authorship networks indicated that the volume of publications does not always align with collaboration influence, as evidenced by the centrality of countries/territories such as France and Saudi Arabia.

Our co-citation network analysis unearthed a tightly knit research community, showing a strong collaborative network among the most-cited papers in eminent journals. This network elucidated how pioneering studies in areas such as e-skin technologies and wearable sensors have considerably shaped ongoing research trajectories. For example, electronic skin technology can continuously monitor physiological signals and enhance patient care [55]. 

Moreover, a cluster analysis based on reference co-citations revealed key thematic clusters, signifying focused explorations into nanotechnology and patient-centered interfaces. These clusters not only mapped out the intellectual structure of the domain but also highlighted emergent themes, such as the integration of advanced materials with HCI in patient care, addressing both present and future healthcare challenges. Especially, nanotechnology can improve healthcare HCI by enabling the development of highly sensitive wearable sensors that monitor vital signs in real time, improving patient care and health management [56].

Our keyword burst analysis reinforced certain terms, such as “electronic health records” and “patient safety”, as foundational. Moreover, the recent rise (2021-2023) in the prominence of keywords such as “artificial intelligence” implies that the HCI–healthcare area tends to transfer to human–AI interaction in healthcare. This may also mean that AI applications will become more frequent in HCI–healthcare academic and industrial areas in the future.

In the future, the domain of HCI in healthcare will undergo further transformation. The advent of sophisticated emerging technologies promises to deepen HCI’s integration into everyday healthcare and foster more personalized and responsive care. In conclusion, our study not only encapsulates the substantial progress within HCI in healthcare but also advocates for sustained interdisciplinary research. By harnessing the strengths of prolific contributors and the power of emerging technologies, HCI is expected to play a pivotal role in elevating the quality of care and improving health outcomes worldwide. 

Although our bibliometric research provides some implications in the HCI healthcare area, several limitations should be addressed in the future. First, although the WoS is a representative and widely used source for bibliometric analysis, WoS has some problems such as the error of DOI, missing author address, regional bias, and under-representation of non-English publications [13,57,58,59]. Thus, future research can employ other sources such as Scopus, IEEE Xplore, or PubMed to provide a better understanding of the HCI healthcare area. Second, future studies could conduct funding analyses to identify the funding agencies that most frequently support the HCI healthcare sector.

## Figures and Tables

**Figure 1 healthcare-12-02467-f001:**
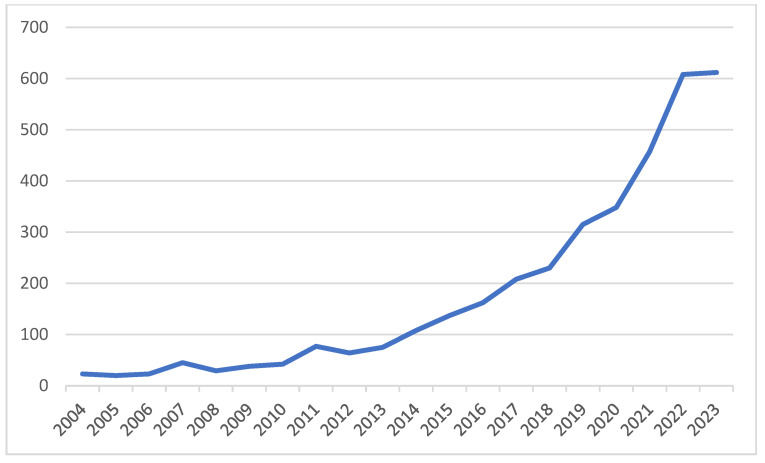
Number of publications per year from 2004 to 2023.

**Figure 2 healthcare-12-02467-f002:**
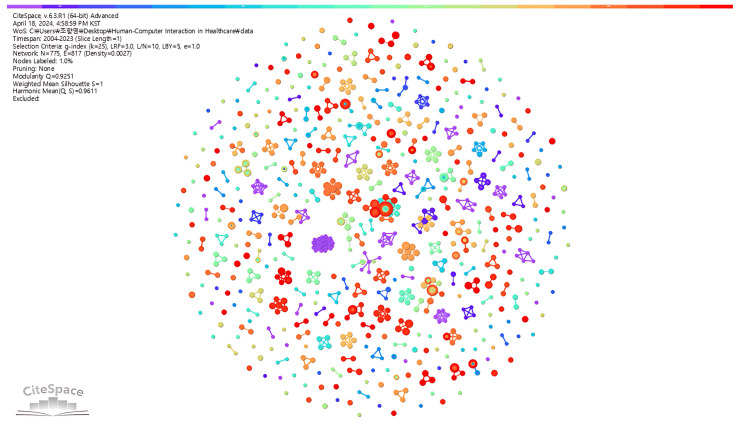
Visualization of co-authorship networks.

**Figure 3 healthcare-12-02467-f003:**
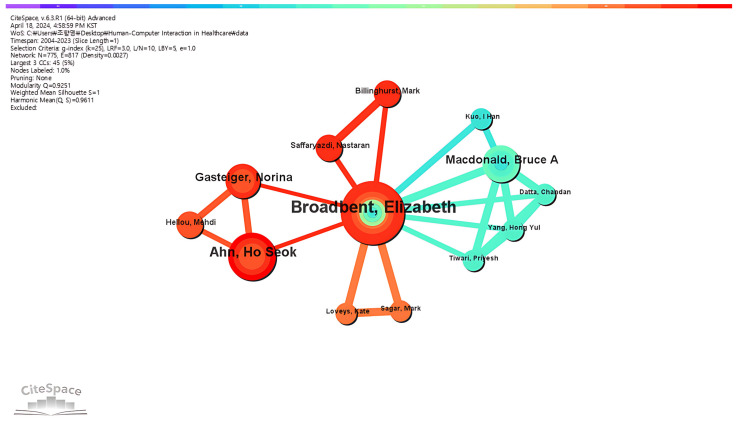
Visualization of the most productive author collaboration network.

**Figure 4 healthcare-12-02467-f004:**
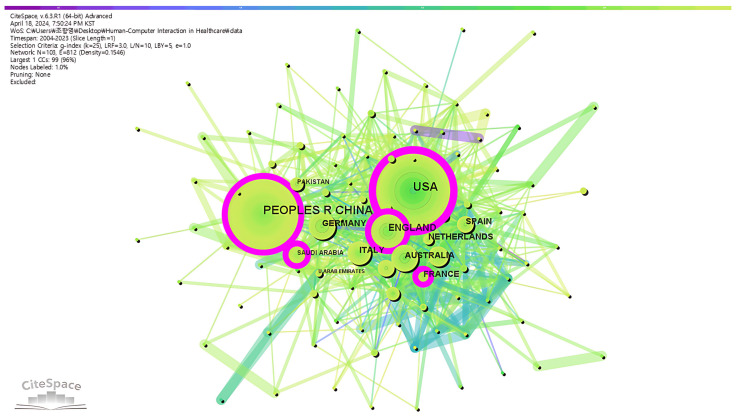
Country/region-based collaboration network.

**Figure 5 healthcare-12-02467-f005:**
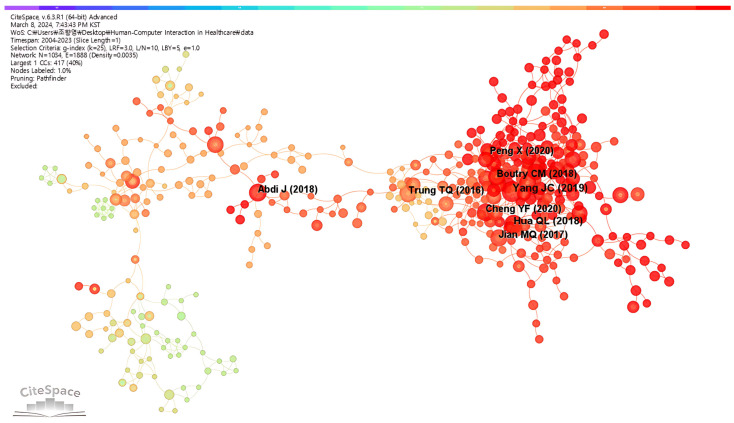
Reference/literature co-citation network [33,34,35,36,37,38,39,40].

**Figure 6 healthcare-12-02467-f006:**
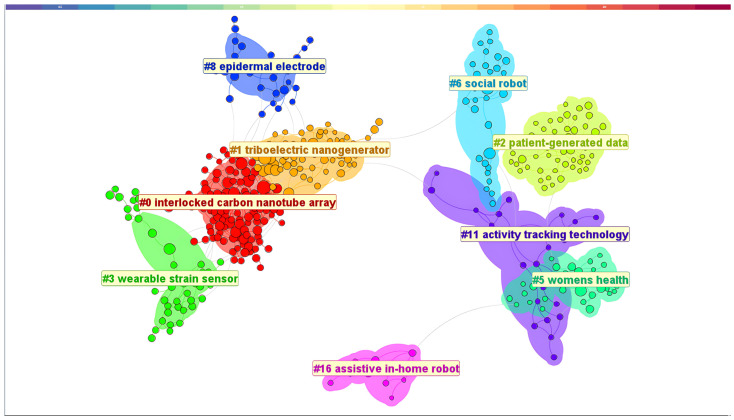
Outcomes of cluster analysis.

**Figure 7 healthcare-12-02467-f007:**
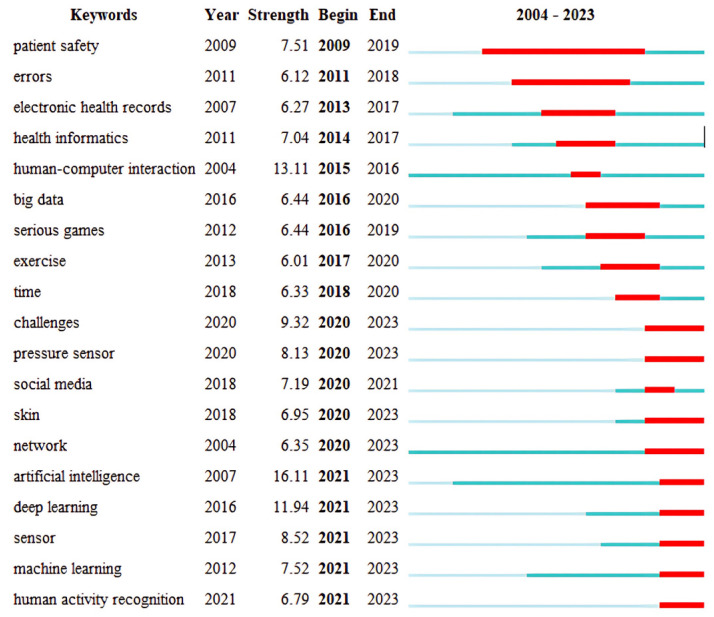
Top 20 keywords with citation bursts.

**Table 1 healthcare-12-02467-t001:** Data extraction process.

Data Source	SCI-E, CPCI-S, SSCI, ESCI, CPCI-SSH, and AHCI of the WoS Databases
Search query	Topic = “human computer interface” OR “human computer communication” OR “human computer interaction” OR “HCI” OR “human robot interaction” OR “human AI interaction” OR “human artificial intelligence interaction” OR “human machine interaction” OR “man machine interaction” or “computer human interaction” AND “health” OR “healthcare” OR “medicine” OR “mental health” OR “behavior health”
Selection criteria	“Article”, “Proceeding Paper”, “Review Article”
	Document language: “English”
	Publication years: 2004–2023
Accessed time	3 March 2024
Number of papers	3598

**Table 2 healthcare-12-02467-t002:** List of the top 10 authors, ranked by their publication output.

Rank	Authors	Number of Papers	Authors’ Affiliations
1	Elizabeth Broadbent	20	The University of Auckland, New Zealand
2	Madeline Balaam	10	KTH Royal Institute of Technology, Sweden
3	Ho Seok Ahn	9	University of Auckland, New Zealand
4	Goldie Nejat	8	University of Toronto, Canada
5	Bruce MacDonald	7	University of Auckland, New Zealand
6	Gavin Doherty	7	Trinity College Dublin, Ireland
7	Eduard Fosch-Villaronga	7	Leiden University, Netherlands
8	Norina Gasteiger	6	University of Auckland, New Zealand
9	Suzanne Bakken	6	Columbia University, United States
10	Dena Al-Thani	6	Hamad Bin Khalifa University, Qatar

**Table 3 healthcare-12-02467-t003:** Top 10 institutes based on the number of published papers.

Rank	Affiliation	Number of Papers
1	University of Washington, United States	71
2	Chinese Academy of Sciences, China	69
3	Georgia Institute of Technology, United States	41
4	University College London, United Kingdom	39
5	Tsinghua University, China	33
6	Northeastern University, United States	31
7	University of Auckland, New Zealand	30
8	University of Chinese Academy of Sciences, China	30
9	Carnegie Mellon University, United States	29
10	Massachusetts Institute of Technology, United States	29

**Table 4 healthcare-12-02467-t004:** Top 10 nations/regions according to the number of publications.

Rank	Country/Region	Number of Papers
1	USA	945
2	People’s Republic of China	721
3	England	322
4	Germany	232
5	Australia	177
6	India	176
7	Canada	175
8	Italy	146
9	Spain	116
10	The Netherlands	114

**Table 5 healthcare-12-02467-t005:** Top 12 countries/territories according to the degree of betweenness centrality.

Rank	Country/Region	Centrality	Number of Papers
1	USA	0.28	945
2	People’s Republic of China	0.16	721
3	England	0.16	322
4	France	0.16	94
5	Saudi Arabia	0.12	65
6	Pakistan	0.09	47
7	Spain	0.08	116
8	United Arab Emirates	0.08	15
9	Italy	0.07	146
10	Germany	0.06	232
11	Australia	0.06	177
12	The Netherlands	0.06	114

**Table 6 healthcare-12-02467-t006:** Top 10 journal/conference proceeding rankings by co-citation count.

Rank	Cited Counts	Journal/Conference Name
1	758	*Lecture Notes in Computer Science*
2	549	*Sensors*
3	488	*PLoS ONE*
4	452	*Journal of Medical Internet Research*
5	399	*Advanced Functional Materials*
6	398	*Advanced Materials*
7	396	*IEEE Access*
8	393	*International Journal of Human–Computer Studies*
9	377	*ACS Applied Materials & Interfaces*
10	363	*Nature*

**Table 7 healthcare-12-02467-t007:** Top eight articles ranked by citation count.

Rank	Cited Count	Reference	Summary	Published Journal/Source
1	31	Yang et al. [33]	Explored advancements in electronic skin (e-skin) technology, highlighting its use in health monitoring and diagnostics.	*Advanced Materials*
2	26	Hua et al. [34]	Introduced a novel, highly stretchable e-skin technology that enhanced sensing capabilities and applied it in smart prostheses for advanced temperature and pressure assessment.	*Nature Communications*
3	23	Trung and Lee [35]	Reviewed flexible and stretchable physical sensors for wearable devices in human activity and healthcare monitoring; discussed the main approaches, applications, and ongoing challenges.	*Advanced Materials*
4	23	Cheng et al. [36]	Devised an MXene-based piezoresistive sensor, mimicking human skin sensitivity through spine-like microstructures; it showed promise for wearable electronics owing to its cost-effectiveness, scalability, high sensitivity, and skin conformity.	*ACS Nano*
5	23	Abdi et al. [37]	Conducted a literature review to explore the applications of socially assistive robots (SARs) in elderly care, identifying five key functions: emotional therapy, cognitive training, social facilitation, companionship, and physical therapy.	*BMJ Open*
6	22	Peng et al. [38]	Developed a breathable, biodegradable, and antibacterial e-skin using all-nanofiber triboelectric nanogenerators, allowing for the real-time and self-powered monitoring of physiological signals and joint movements; they presented a novel approach for multifunctional e-skins with practical utility.	*Science Advances*
7	22	Boutry et al. [39]	Developed a soft e-skin with capacitors to detect real-time forces; it features pyramid microstructures for improved sensitivity and stability, provides tactile feedback for robot arm control, and shows promise in robotics.	*Science Robotics*
8	21	Jian et al. [40]	Introduced a simple method for fabricating flexible and highly sensitive pressure sensors using biomimetic hierarchical structures and highly conductive active membranes. It offers potential applications in wearable electronics and human–machine interfaces.	*Advanced Functional Materials*

**Table 8 healthcare-12-02467-t008:** Major clusters.

Cluster ID	Silhouette	Size	Label (LLR)	Label (LSI)
0	0.943	117	Interlocked carbon-nanotube array	Flexible pressure sensor
1	0.922	68	Triboelectric nanogenerator	Human–machine interaction
2	0.99	55	Patient-generated data	Patient-generated data
3	0.968	40	Wearable strain sensor	Flexible strain sensor
5	0.917	35	Women’s health	Women’s health

Note: “Size” refers to the number of cited publications in a cluster. “Silhouette” denotes the homogeneity of a cluster [45].

## Data Availability

The data presented in this study are available upon request from the corresponding author.

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
