# Peer review of "Human–Computer Interaction in Healthcare: A Bibliometric Analysis with CiteSpace"

_healthcare, 2024, doi:10.3390/healthcare12232467_

Round 1

Reviewer 1 Report

Comments and Suggestions for Authors

See the attachment

Comments on the Quality of English Language

should be polished

Author Response

Thank you again for your valuable suggestions. Thanks to your valuable guidelines, we believe that our study is well improved. If there are any questions about our revision, please do not hesitate to tell us. We will try our best to meet your requirements!

Reviewer 2 Report

Comments and Suggestions for Authors

This publication reports about the results of  bibliometric analysis about the  landscape of HCI within the healthcare sector. Your analysis used Citespace as a review tool. It would be beneficial for your paper to report about the state of the art of bibliometric literature review tools. There are many other tools that can be used for such purpose e.g. VOSviewer , BibExcel, HistCite, Pajek, Netdraw, and Publish or Perish. Or even just excel for basic bibliometric analyses.

Author Response

(The authors gave the same response as above.)

Reviewer 3 Report

Comments and Suggestions for Authors

The manuscript "Human-Computer Interaction in Healhcare: A Bibliometric Analysis with CiteSpace" explores the growing role of Human-Computer Interaction (HCI) in healthcare, focusing on how it can improve medical services, enhance patient experiences, and support healthcare professionals. As HCI research in healthcare is expanding, the study aims to provide a comprehensive overview of trends, key contributors, and emerging topics within this field through a bibliometric analysis. Using CiteSpace, a tool for visualizing scientific literature, the authors analyze thousands of publications from 2004 to 2023 to identify leading researchers, institutions, countries, and influential research topics. By doing so, the paper offers valuable insights into the current landscape of healthcare HCI, guiding future research and innovation in areas such as wearable technology, electronic health records, and patient-centered applications.

  • The introduction effectively captures the significance of Human-Computer Interaction (HCI) within the healthcare sector. However, it could benefit from a clearer explanation of the gaps in current research that this paper addresses, specifically how it contributes beyond existing bibliometric analyses in healthcare HCI.
  • The paper demonstrates a thorough data collection approach from credible sources (e.g., Web of Science databases) and employs CiteSpace well for bibliometric analysis. However, the manuscript could further clarify the criteria for selecting the 3,598 papers analyzed, potentially providing a brief explanation of the chosen time frame and why it spans from 2004 to 2023.
  • The methodology section could include more information about the limitations of using only Web of Science and whether any potential biases from excluding other databases were considered.
  • The authors provide an in-depth co-authorship and institutional analysis, identifying the key contributors and collaboration networks within healthcare HCI research. One recommendation would be to elaborate on the implications of these collaborative networks, such as how they may influence the direction and quality of research outputs in this field.
  • The study identifies top-performing authors, institutions, and countries. Adding a discussion on how institutional or regional policies and funding might impact research output could enhance the depth of analysis.
  • The cluster analysis on research themes is well-conducted, with clear labels and insights into the primary research themes in healthcare HCI, such as wearable technology and electronic skin. Further exploration of each theme’s practical applications or current real-world examples could improve readers' understanding of how these findings translate into healthcare advancements.

However, there are some potential shortcomings in the manuscript that could be addressed to improve its clarity and impact:

  • The study relies exclusively on the Web of Science for data collection. This approach might overlook relevant research in healthcare HCI published in other databases like Scopus, IEEE Xplore, or PubMed, potentially limiting the comprehensiveness of the findings.
  • The methodology section could benefit from a more detailed description of the criteria for data selection, including an explanation for the chosen time frame (2004–2023). Additionally, the reasoning behind using specific keywords and the potential for missing alternate terminologies in HCI could be addressed
  • Expanding on the role of thematic clusters in advancing specific areas of healthcare (e.g., patient safety, AI-driven healthcare interfaces) could add valuable insights into the future directions of healthcare HCI.

Comments on the Quality of English Language

Moderate editing required.

Author Response

(The authors gave the same response as above.)

Round 2

Reviewer 1 Report

Comments and Suggestions for Authors

see the attachment

Comments on the Quality of English Language

should be polished

Author Response

(The authors gave the same response as above.)
